# Biochemical Properties Affecting the Nutritional Quality, Safety, and Aroma of Dry-Cured Products Manufactured from Meat of Rare Native Pig Breeds

**DOI:** 10.3390/foods10071597

**Published:** 2021-07-09

**Authors:** Ewelina Węsierska, Joanna Sobolewska-Zielińska, Małgorzata Pasternak, Katarzyna Niemczyńska-Wróbel, Robert Gąsior, Krzysztof Wojtycza, Henryk Pustkowiak, Iwona Duda, Władysław Migdał

**Affiliations:** 1Department of Animal Product Technology, University of Agriculture in Krakow, Balicka 122, 30-149 Krakow, Poland; malgorzata.pasternak@urk.edu.pl (M.P.); katarzyna.niemczynska@urk.edu.pl (K.N.-W.); iwona.duda@urk.edu.pl (I.D.); wladyslaw.migdal@urk.edu.pl (W.M.); 2Department of Food Analysis and Evaluation of Food Quality, University of Agriculture in Krakow, Balicka 122, 30-149 Krakow, Poland; joanna.sobolewska-zielinska@urk.edu.pl; 3Central Laboratory, National Research Institute of Animal Production, Balice, 32-083 Krakow, Poland; robert.gasior@izoo.krakow.pl (R.G.); krzysztof.wojtycza@izoo.krakow.pl (K.W.); 4Department of Cattle Breeding, University of Agriculture in Krakow, Al. Mickiewicza 24/28, 30-059 Krakow, Poland; henryk.pustkowiak@urk.edu.pl

**Keywords:** Pulawska, Zlotnicka Spotted meat, dry-cured products, maturing, quality, safety

## Abstract

The aim of study was to compare the biochemical properties affecting the nutritional quality, safety, and aroma of dry-cured products manufactured from valuable meat of rare native pig breeds: Pulawska (Pul) and Zlotnicka Spotted (ZS). The count of lactic acid bacteria (4.4 log cfu/g) and the release of palmitic (23.1% and 25.9%), oleic (44.1% and 42.2%), and linoleic acids (8.3% and 7.8%), as well as arginine (30.0 and 44.3 mg/kg), histidine (25.8 and 20.6 mg/kg), and lysine (26.8–22.9 mg/kg), shaped the final pH (5.3 and 5.4) in Pul and ZS products during the 4 week maturing, respectively. Lastly, Pul and ZS meat differed in the proportion of decanoic, lauric, stearic, arachidic, and conjugated linoleic acids. The high content of putrescine (23.7 mg/kg), cadaverine (54.3 mg/kg), and tyramine (57.2 mg/kg), as well as a twofold greater share of histamine (163.2 mg/kg) and tryptamine (9.1 mg/kg), indicated a more advanced decarboxylation of ZS meat. Volatile compounds differentiating Pul and ZS meat were primarily hexanal, 3-hydroxybutan-2-one, phenylacetalaldehyde, 2,3-dimethyl-2-cyclopenten-1-one, 2-cyclopenten-1-one, and 3-methyl- and 2-cyclopenten-1-one. Most marked volatile compounds were obtained as a result of microbial activity (acetic acid, 3-methylbutan-1-ol, ethanol, acetone, and 3-hydroxybutan-2-one), advanced lipid oxidation, and decomposition of secondary oxidation products (hexanal, phenylacetaldehyde, and 2-cyclopenten-1-one).

## 1. Introduction

The pork of native breeds Pulawska (Pul) and Zlotnicka Spotted (ZS), due to the appropriate proportion of water (69.8–74.0% and 69.2–73.6%, respectively) to other basic chemical components and sufficient drip loss (1.7–4.7% and 2.4–3.4%, respectively), is beneficial for manufacturing of regional products of a unique culinary value. For comparison, the commercially used pigs give incomparable drip loss values of 3.0–4.3% [1,2,3,4,5,6]. The studies of Kapelański et al. [7] and Grześkowiak et al. [8] revealed PSE (pale, soft, exudative) and DFD (dark, firm, dry) defects only for single Pul and ZS carcasses. These observations confirmed the correct course of post-slaughter maturation, suitably low pH values, and technological usefulness of the meat. The quality of Pul and ZS meat does not differ significantly in terms of pH, brightness, and cutting force, but there are some differences in the chemical composition [9]. Slightly different marbling creates not only an individual sensory appearance and attractiveness of slices on the cross-section, but also a juiciness and a distinct flavor of Pul and ZS meat [7,8]. The high content of protein shapes the nutritional value and texture different to industrial breed meat [10]. Meat of the Pul breed is characterized mainly by special organoleptic value and is suitable for the production of regional slow food. The high quality of ZS pork allows it to be used for the production of cooked and baked hams and sirloins [1,7,9,11,12,13,14]. The meat of Pulawska and Zlotnicka White and Spotted breeds of pigs is a good-quality raw material used for production of traditional and regional smoked meat products gaining high sensory scores and good recognition among consumers [15,16]. For this reason, the production of dry-cured, long-maturing meat products is technologically attractive to better understand the raw material and the transformations that it undergoes. The European Food Information Resource Network (EuroFIR) indicates that traditional products have specific features that make it possible to distinguish them from other similar products. Traditional products of animal origin should have a traditional recipe and a traditional method of production, while they should be made from traditional raw materials, preferably using native animal breeds. A similar direction of meat production from native breeds can be found in southern European Union countries. The free-range system increases the value of animal products due to the influence of outdoor rearing on the chemical, physical, and organoleptic characteristics of the product. Traditional products made from pork of native breeds are highly appreciated: Spanish “Ibérico” ham, Tuscan “Cinta senese”, Basque ham, and the Corsican prisuttu are produced from Iberian, Cinta Senese, Pie Noir du pays Basque, and Corsican Black or Spotted native pig meat, respectively. Long-term maturing is done in appropriate climatic conditions, at a temperature of 15–18 °C and a relative humidity of 85–90%, lowered to 70–80% at the further stage of maturing. The temperature of the curing smoke at the time of cold smoking which lasts for several days should not be higher than 18–25 °C [17]. Long-term maturing smoked products constitute an important segment of regional products in Italy, Spain, Portugal, France, and recently even the Czech Republic.

The aim of the study was to compare the progress of 4 week maturation of dry-cured ham made from the meat of two native breeds of pigs (Pulawska and Zlotnicka Spotted), spontaneously fermented using the same method. The progress of drying was assessed on the basis of changes in the chemical composition (water, protein, fat, ash, salt content), pH, and water activity, as well as the succession of acidifying and denitrifying microflora. The advancement of the proteolysis, amino-acid decarboxylation, lipolysis, and fat oxidation was studied by measuring the share of amino acids, amine groups, biogenic amines, fatty-acid content, and TBA index, respectively. Volatile compounds were analyzed to illustrate the advancement of microbial fermentation, amino-acid and fatty-acid catabolism, and lipid oxidation. Since meat from the native breeds is an excellent raw material for the production of slow food, the analyses planned in this study were carried out during the 4 week ripening of dry-cured hams, low-processed and produced using the same traditional method with the complete absence of advanced technologies, including heat treatment.

## 2. Materials and Methods

### 2.1. Dry-Cured Product Manufacturing

The raw material was the topside part of ham muscles after the group of pelvic bones group was removed, taken from pigs of native breeds Pulawska (Pul) and Zlotnicka Spotted (ZS). Thirty fatteners of each of the discussed breeds were chosen. The pigs were fed ad libitum with a complete feeding mixture to obtain a body weight of 125–130 kg. Three times, after reaching the slaughter mass, pigs were slaughtered (the cuts for production were randomly selected from a pool of 30 carcasses). The meat was obtained from the same litter of pigs (half-siblings). All activities related to slaughter and post-slaughter processing were carried out in an industrial slaughterhouse in accordance with applicable regulations. The weight of chilled ham after boning varied between 1.2 and 1.3 kg. To obtain a salt content of no more than 4.5% in the ready-to-eat products, salt was added during dry curing in the amount of 2% of non-iodized salt and 1% of curing salt (min. 98.4% NaCl, 0.6% NaNO_2_, Anna, Krakow, Poland) related to the weight of the raw meat. Additionally, 0.3% sugar and spices (0.9% juniper berries, 0.4% allspice, 0.4% pepper, 0.4% garlic) were added in relation to the meat weight. The ham was placed in containers for curing in dry (2 weeks) and then wet conditions (1 week) at 4–7 °C. The ratio of the amount of the brine to the meat was 2:3. The 8° brine contained 87 g of NaCl per 1000 mL of water. The cooled cure was slightly boiled beforehand to eliminate any accidental microflora. Ham was smoked cold three times (15–25 °C) with alder-beech chips in daily intervals (8 h/day). Ripening at a temperature of 10–12 °C and 85–90% relative humidity (first week) and 85% (subsequent time of maturing) was carried out in a climate chamber KK350 TOP+ (POL-EKO) for 4 weeks. The slaughter was carried out three times. The obtained carcasses were used each time for three batches of ham production. Three whole dry-cured hams (three Pul and three ZS products) as replications in each batch and three samples from each ham were analyzed.

### 2.2. Sampling

Samples of the raw material were collected just after dry salting and the smoking procedures (ripening period: 0). The cuts were also sampled at different times throughout the maturing (after second and four weeks). The material for analysis was cut into 1.5 cm thick slices. The first one (external) was rejected because of its dried surface and influential water loss. After separating, the samples for the basic chemical composition (water, protein, fat, ash, salt), pH, water activity, and the total count of microorganisms, as well as the counts of acidifying and denitrifying microflora, were estimated as the process of maturing progressed. The contents of free amino acids, free amine groups, and biogenic amines, as well as the profiles of fatty acids and volatile compounds, were determined only in the dry-cured ham ready for consumption, after 4 weeks of maturing. All meat samples for analysis were transported to the appropriate laboratories under refrigerated conditions. Analyses were done immediately after delivery. The test samples were prepared in accordance with the appropriate standards.

### 2.3. Analysis

#### 2.3.1. Basic Chemical Composition and Physicochemical and Microbiological Characteristics

The basic chemical composition was evaluated according to the Polish and European recognized standards. The moisture and ash contents were determined by drying samples to their stable weight [18]. The protein content was determined with the use of the Kjeldahl method with the set-type 322 (Büchi, Flawil, Switzerland) using a 6.25 factor [19]. Free fat was evaluated with the use of the Soxhlet method with ethyl ether extraction [20] and the chloride content with the use of Volhard’s method [21]. The energy value of meats was determined by the Atwater specific factors of Food and Agriculture Organization of the United Nations (2003). The pH was measured with pH-meter type CP-411 and electrode type PP-3 (Elmetron, Zabrze, Poland) in water homogenate (meat–water ratio, 1:3) and water activity was determined with the Lab Master (Novasina, Lachen, Switzerland), following the producer manual. Analyses for bacteriological examination were done for the following: the total count of microorganisms in mesophilic conditions (Standard Methods Agar, Biomérieux, 30 °C/72 h) [22], acidifying microflora (MRS Agar, Biomérieux, acetic acid used for pH reducing to 5.4, 30 °C/24–48 h, anaerobic chamber with a 20% CO_2_ enriched atmosphere (Sheldon Manufacturing Inc., Cornelius, OR, USA) [23], and denitrifying microflora (Baird Parker Agar Base, yolk emulsion and sodium tellurite, Biomérieux; coagulase-negative cocci classified on the basis of bound coagulase and free coagulase activity, 37 °C/24 h) [24].

#### 2.3.2. Proteolytic Changes and Biogenic Amine Content

Free amino groups were evaluated in water and in phosphotungstic acid [25]. Water-soluble nitrogen (WSN) was determined in the water-soluble fraction (WSF) of the sample. The contents of free amino-acid groups in water (WSF) and the phosphotungstic acid (PTA)-soluble fractions were measured using a 2,4,6-trinitrobenzenesulfonic acid (TNBS) (Sigma-Aldrich, Saint Louis, MI, USA). A phosphotungstic acid soluble fraction of meat sample was prepared from WSF. The WNS fraction was used to determine the free amino acids as follows: 5 mL of the water-soluble protein fraction was added to 5 mL of 40% trichloroacetic acid (TCA) and stirred on a Vortex RS-va10 (Phoenix Instrument, Garbsen, Germany) for 10 min, before centrifuging at 12,000× *g* for 10 min in a precooled centrifuge MPW-352R, REF (MPW Med. Instruments, Warszawa, Poland) at 4 °C. Then, 2 mL of solution was removed and dried under nitrogen. The obtained residue was dissolved in 1 mL of 20 mM HCl. The chromatographic separation (HPLC) of free amino acids was performed with Dionex UltiMate 3000 (Thermo Scientific, Waltham, MA, USA). Separation was carried in a Nova-Pak reverse-phase C18 column, 4 µm particle size, 150 × 3.9 mm (Waters, Milford, MA, USA), thermostated at 37 °C. The two solvent reservoirs contained the following eluents: (A) acetate–phosphate buffer pH = 5.2 and (B) a mixture of acetonitrile and water (60:40, *v*/*v*) according to the Waters procedure. The elution program consisted of a gradient system with a flow rate of 1 mL·min^−1^ (100% A for 0.5 min, 98% A and 2% B for 14.5 min, 93% A and 7% B to 19 min, 90% A and 10% B to 32 min, 67% A and 33% B to 33 min, 100% B to 37 min, and 100% A to 64 min). Fluorometric detection was carried out using excitation and emission wavelengths of 250 nm and 395 nm, respectively. Analytical kit ACCQ Tag (Waters, USA) was used as a reference.

The biogenic amine analysis was performed using HPLC according to the following method: a 10 g sample of meat was mixed with 15 mL of 6% trichloroacetic acid (TCA), homogenized for 2 min (1000 rpm, Vortex), and centrifuged at 14,000× *g* for 20 min at 4 °C (MPW Med. Instruments, Warszawa, Poland). The obtained supernatant was collected and filtered through Whatman No. 1 filter paper, and the residue was reextracted with 15 mL of fresh of 6% TCA. The volume of the combined supernatants was completed to 50 mL by 6% TCA. The derivatization process was performed by mixing 1 mL of extract with 1 mL of dansyl chloride acetone solution (10 mg·mL^−1^) and 0.5 mL of saturated NaHCO_3_ solution. Incubation was carried out at 40 °C for 60 min with occasional shaking in a thermoblock (Dry Block Heater, IKA, Staufen, Germany). The biogenic amines were extracted twice using 1 mL of diethyl ether for 10 min (22 °C). The extracts were dried in a stream of nitrogen, and the residue was dissolved in 1 mL of acetonitrile. The solution standards of biogenic amines were prepared by mixing 1 mL of each free base standard solution, containing 0.1 mg·mL^−1^ of each biogenic amine with 5 mL of dansyl chloride acetone solution. The derivatization and extraction procedure was performed in the same manner as the samples. The chromatographic separation of free amino acids was performed using Dionex UltiMate 3000 HPLC apparatus (Thermo Scientific, Waltham, MA, USA). The separation was carried in a Nova-Pak reverse-phase C18 column, 4 µm particle size, 150 × 3.9 mm (Waters, Milford, MA, USA), thermostated at 30 °C. The two solvent reservoirs contained the following eluents: (A) acetonitrile and (B) HPLC-grade water at a flow rate of 0.8 mL·min^−1^. The elution program was: 65% A and 35% B for 1 min, increasing to 80% A and 20% B for 9 min, increasing to 90% A and 10% for 2 min, increasing to 95% A and 5% B for 4 min, holding for 7 min, and returning to 65% A and 35% B, before holding for 5 min. Fluorometric detection was carried out using excitation and emission wavelengths of 340 nm and 530 nm, respectively. The solution of standards of eight biogenic amines was prepared in 0.1 M HCl in a concentration of 1 mg·mL^−1^ of every amine; each of the following reference substances was dissolved in 10 mL of 0.1 M HCl: tyramine hydrochloride (Sigma-Aldrich, Buchs, Switzerland)—12.7 mg, tryptamine hydrochloride (Sigma-Aldrich, Saint Louis, MI, USA)—12.3 mg, histamine dihydrochloride (Sigma-Aldrich, Beijing, China)—16.6 mg, putrescine dihydrochloride (Sigma-Aldrich, Buchs, Switzerland)—18.3 mg, cadaverine dihydrochloride (Sigma-Aldrich, Buchs, Switzerland)—17.1 mg, spermine tetrahydrochloride (Sigma-Aldrich, Buchs, Switzerland)—17.2 mg, spermidine trihydrochloride (Sigma-Aldrich, Buchs, Switzerland)—17.5 mg, and 2-phenylethylamine hydrochloride (Sigma-Aldrich, Tokyo, Japan)—13.0 mg. Prepared solution standards were then 100-fold diluted. All reagents used were HPLC-grade (Sigma-Aldrich, Saint Louis, MI, USA).

#### 2.3.3. Fatty-Acid Profile and Oxidative Changes

The fatty-acid profile was determined by gas chromatography from the fat extracted from the samples of ready-to-eat products. Lipids were extracted from samples with a chloroform–methanol (2:1) mixture, according to the method of Folch, Lees, and Sloane [26]. Trace GC Ultra (Thermo Electron Corporation, Rodano, Italy) with a column Supelcowax 10 (Sigma-Aldrich, Saint Louis, MI, USA) 30 m × 0.25 mm × 0.25 μm was used for the determination. The analysis was carried out under the following conditions: helium carrier gas 1 mL·min^−1^, split flow 10 mL·min^−1^, injector temperature 220 °C, detector temperature 250 °C, and starting temperature of the column (first 3 min) 160 °C, increased by 3 °C min^−1^ to 210 °C and maintained for 25 min. Individual fatty-acid methyl esters were identified by comparison to the standard mixture (Supelco 37 Component FAME Mix, Sigma-Aldrich, USA). The thiobarbituric acid (TBA) index was determined by the spectrophotometric method in a Helios γ spectrophotometer (Thermo Electron Corporation, Beverly, MA, USA) and expressed as mg of malondialdehyde (MDA) in kg of meat. Furthermore, 10 g of sample was homogenized with 34.25 mL of cold 4% perchloric acid and 0.75 mL of alcoholic butylated hydroxytoluene solution (4000 rpm for 2 min in a precooled centrifuge (MPW Med. Instruments, Warszawa, Poland)). Then, 5 mL of the filtrate was combined with 5 mL of 0.02 M TBA and heated for 1 h. Absorbance was read at 530 nm. The blank was 5 mL of 4% perchloric acid solution and 5 mL of TBA [27].

#### 2.3.4. Volatile Compound Content

Volatile compounds were extracted and analyzed using a gas chromatograph–mass spectrometer GCMS-QP 2010 Plus (Shimadzu, Duisburg, Germany), with a 50/30 μm DVB/CAR/PDMS fiber (Supelco, Bellefonte, PA, USA) and Zebron ZB-5MSi and ZB-Wax columns 30 m × 0.25 mm × 0.25 μm (Zebron, Phenomenex^®^, Torrance, CA, USA) [28]. The identification was done using mass spectral libraries (NIST08, NIST08s, and FF NSC1.3) and retention indices RI, on the basis of analyses of *n*-paraffins (Supelco, Merck Group, Poznan, Poland), compared with values from National Institute of Standards and Technology. Some of the compounds were verified by standards (S); for quantitation, methyl caproate (internal standard) was used (Sigma-Aldrich, Saint Louis, MI, USA).

#### 2.3.5. Statistical Analysis

The statistical analysis was performed using the Statistica software for Windows, version 13.3 (Statsoft, Krakow, Polska). The effect of the ripening time on the chemical properties and microbiological quality was tested using one-factor (ripening period: 0, 2, 4 weeks) and two-factor (ripening period: 0, 2, 4 weeks; breed: Pul, ZS) analyses of variance (ANOVA with fixed and orthogonal factors). Duncan post hoc tests were used to compare the means at *p* < 0.05.

## 3. Results and Discussion

### 3.1. Basic Chemical Composition, and Physicochemical and Microbiological Characteristics

Despite the removal of the external fat cover of different thickness for both breeds during the boning, Pul and ZS dry-cured products differed in their basic chemical composition. To shorten the presentation, the results are presented in Table 1 as arithmetic means obtained after a two-factor analysis of variance. Pul products had a lower protein (18.8%) and a higher fat content (5.0%) compared to ZS products (22.6% and 4.5%, respectively). According to the differences in the intramuscular fat content of Pul and ZS products, the energy values were calculated as 157.8 and 149.6 kcal, respectively. The basic chemical composition and physicochemical parameters of Pul and ZS products changed significantly over the 4 week maturing. The loss of water from 74.3% to 69.1% (*p* < 0.05) resulted in an increase in total protein (from 18.2% to 20.8%) (*p* < 0.05) and fat content (from 3.4% to 6.4%) (*p* < 0.05). Changes in the basic chemical composition are of great importance for the organoleptic quality and nutritional value of dry-cured products manufactured by both traditional and industrial methods. The share of water, protein, fat, and salt determines the quality properties such as water-holding capacity, juiciness, texture, color, and overall taste during maturing, as confirmed by [15,29,30,31]. The quantitative and qualitative composition of the microflora of raw sausages varies and depends on accidental contamination of the meat. In the initial period of maturation, there is a cenobiotic exchange of microflora from accidental to technologically desirable, i.e., acidifying, denitrifying, and aromatizing. During the initial maturation and cold smoking, the dominant microflora were cocci and rod-shaped bacteria, which lowered the pH and intensified the color of the meat. In the further stage of maturation and during the post-production maturation, staphylococci and micrococci shaped the flavor profile of the meat. Maturation time significantly affected the analyzed total count of microorganisms (*p* < 0.05), technologically desirable microflora (*p* < 0.05) (acidifying microflora (lactic acid bacteria) and denitrifying microflora (coagulase-negative cocci)), and the final pH of Pul and ZS products (*p* < 0.05) (Figure 1). Despite the higher average salt content (1.9%) and the lower pH (5.5), the total count of microorganisms (4.3 log cfu/g) and denitrifying bacteria (2.7 log cfu/g) of Pul was more than that of ZS products. The increased salinity and acidity of dry-cured meat products as a result of maturing and a reduction in the water activity does not inhibit the development of halophilic denitrifying and proteolytic microflora [32,33,34,35]. Staphylococcaceae and Micrococcaceae support the process of color, aroma, and safety formation [29]. An increase in the cocci population observed after the second week of analyzed maturing corresponded to a scenario whereby the fermentation of saccharides (the development of acidifying bacteria) acted most intensively during the first 3–5 days, while the proper development of denitrifying microflora occurred a few weeks later, depending on the type of meat product [36,37].

### 3.2. Proteolytic Changes and Biogenic Amines Content

The ready-to-eat Pul and ZS dry-cured products differed significantly in the content of amino acids released by proteolysis, especially alanine (47.4 and 30.4 mg/kg), glutamine (44.8 and 23.2 mg/kg), threonine (43.7 and 31.8 mg/kg), and leucine (40.9 and 32.8 mg/kg) (Table 2). The amount of free amino groups dissolved in water determined the majority of typical proteolysis products (0.6 and 0.7 g/100 g (*p* < 0.05), respectively). Amino groups dissolved in PTA (0.06 g/100 g) in Pul and ZS products proved the presence of the low-molecular-weight fraction of singular amino acids, as well as di- and tripeptides [25,31]. Similar changes in the proteolysis activity and the accumulation of free amino acids during ripening were described in fermented sausages [38], dry-cured ham [39], and Chinese traditional dry-cured loin [40]. Changes in the profiles of metabolites originating from protein degradation affect the sensory quality and safety of dry-cured products during ripening and are still the focus of science [15]. The high proportion of putrescine, cadaverine, and tyramine in ZS (23.7, 54.3, and 57.2 mg/kg) compared to Pul products (4.0, 28.5, and 11.1 mg/kg) confirmed the differences in proteolysis and decarboxylation of ornithine, lysine, and tyrosine, respectively (*p* < 0.05) (Table 3). A nearly twofold greater share of histamine and tryptamine (163.2 and 9.1 mg/kg, respectively (*p* < 0.05)) was determined in the ZS products as the result of decarboxylation of histidine and tryptophan. The production of biogenic amines confirmed the metabolic activity of the microflora [41]. The ratio of protein to fat and a slightly different water content could create an environment more or less conducive to the decarboxylation of amino acids by bacteria. The contents of cadaverine, histamine, and tyramine in dry-cured pork shoulder ripened for 3 months remained at a lower level of 7.5, 6.5, and 21.9 mg/kg, respectively [36]. Manufacturing of dry-cured ham is a safe process, although this meat product can occasionally support the accumulation of amino-acid decarboxylation products. In fact, a high proteolytic activity during maturing favors the decarboxylase activity of many microorganisms. So far, no cases of poisoning with biogenic amines after ingestion of dry-cured ham have been confirmed in the literature. The level of amines is not limited by food safety criteria for maturing meat products. The study of Landeta et al. [42] revealed that the production of biogenic amines was not a widely distributed property among staphylococci. Only 3.6% strains of coagulase-negative staphylococci isolated during industrial Spanish dry-cured ham processes could produce some biogenic amines, including histamine, putrescine, cadaverine, and tyramine.

The levels of these amines showed a time-dependent increase during aging (decrease in moisture and water activity, along with a minimal increase in pH and non-protein nitrogen and free amino-acid content), especially in ZS hams, in line with the studies of Lorenzo et al. [43], Martuscelli et al. [44], Stadnik and Dolatowski [45], and Virgili et al. [46].

### 3.3. Fatty-Acid Profile, Oxidative Changes, and Volatile Compound Content

The dry-cured meat products of Pul and ZS breeds differed in the proportion of selected saturated fatty acids (SFAs): decanoic (C10;0), lauric (C12;0), stearic (C:18;0), and arachidic (C20;0) (*p* < 0.05) (Table 4). The sum of SFAs in ready-to-eat Pul and ZS products was determined at 34.9% and 38.5%, respectively. Only the content of heptadecenoic acid (C17;1) (*p* < 0.05) differed in Pul and ZS products, while the sum of MUFAs in both meats amounted to 54.1% and 50.9%, respectively. There were no significant differences in the sum of ω-6 acids in Pul and ZS products. The only acid differentiating Pul and ZS in the ω-3 group was CLA (*p* < 0.05). In both cases, the sum of ω-3 acids was 0.8%. The quantitative and qualitative composition of fatty acids depends on the composition of the feed mixtures. Pul and ZS pigs were fed the same feed, but their behavior and physiological conditions for fat storage are different. This can change the proportion of fatty acids in meat products. Debrecéni et al. [9], Cebulska [14], and Migdał et al. [15], indicated statistically significant differences in the contents of palmitic acid, oleic acid, LA, ALA, GLA, AA, and DHA in Pul and ZS meat. Probably due to the antioxidative activity of wood pyrolysis, spices, and lactic acid microflora, the TBA index was determined on a low level of 0.3 mg/kg (Pul) and 0.5 mg/kg (ZS) (*p* < 0.05). Low TBA values may also indicate a very good physicochemical and organoleptic quality of the fat, as well as proper cooling of the carcasses after slaughter [37]. The lipolysis directly resulted in the formation of aroma compounds by liberating fatty acids [33]. MDA enables further transformations such as condensation, reactions with released amino acids, and decomposition via the following catalysts: *Pediococcus acidilactici*, *Lactobacillus plantarum*, and *Staphylococcus carnosus* [29]. Spaziani et al. [32], Ravyts et al. [34] and Berardo et al. [35] indicated aldehydes, ketones, and alcohols as secondary products of hydrolytic rancidity and autooxidation of UFAs. Some of them could be the products of microbial metabolism, including carbohydrate fermentation, as well as FAA catabolism or esterification. Longer chains of fatty acids (C14–C18) could become, along with products of proteolysis, the precursors of selected VOCs. According to Olivares et al. [29] and Węsierska et al. [37], the substances obtained in the enzymatic (20%) and nonenzymatic oxidation of UFAs (80%) become the basis for aroma precursors. Volatile compounds developed from autooxidation of lipids give a rancid profile. However, there was no high content of compounds that would adversely change the sensory profile of the Pul and ZS products. The cause could have been the antioxidative activity of phenols supplied during smoking and absorbed by the surface.

There were 48 volatile compounds (VCs) determined by HS-SPME–GC/MS (Table 5). Substances with the largest amount were obtained through biochemical and microbiological transformations of meat tissues (16 VCs) or came from spices (25 VCs). Some of them were the products of pyrolysis of wood (seven VCs). The compounds that differentiated Pul and ZS products (*p* < 0.05) were primarily hexanal, 3-hydroxybutan-2-one (acetoin), phenylacetalaldehyde, 2,3-dimethyl-2-cyclopenten-1-one, 2-cyclopenten-1-one, 3-methyl-, 2-cyclopenten-1-one, and 2-methyl- and acetic acid, obtained as a result of microbial activity (fermentation) and lipid oxidation. The compounds that developed the meat aroma in all meats were hexanal, 3-hydroxybutan-2-one, 3-methylbutanal, and 2-methylcyclopentanone. The main sources of sulfur compounds and terpenes were garlic, pepper, and pyrolysis of lignin with their terpy, herbaceous, woody nuances. Some of them occurred in large quantities in Pul products perhaps due to the lower content of protein and higher fat content. The examples of compounds of spices, as well as of pyrolysis origin, that differentiated Pul and ZS products were 2-furanmethanol, 2-furanmethanol, 5-methyl, eugenol, o-xylene, caryophyllene, cresol, diallyl disulfide, camphor, linalool, guaiacol, γ-terpinene, limonene, and 3-carene. The presence of hexanal (11.3 and 42.5 ng/g), phenylacetaldehyde (20.5 and 8.8 ng/g), 2-cyclopenten-1-one (7.2 and 5.9 ng/g), 1-hydroxypropan-2-one (5.9 and 4.9 ng/g), and 2-methylcyclopentanone (11.7 and 11.1 ng/g) could indicate an increased activity of secondary fat oxidation products despite the small values of the TBA index (0.3 and 0.5 mg/kg) in Pul and ZS products, respectively. The amounts of products of saccharide fermentation such as acetic acid, 3-methylbutan-1-ol, ethanol, acetone, and 3-hydroxybutan-2-one confirmed a high enzymatic activity of microorganisms [47,48]. These volatile compounds are mostly in line with those reported by different authors for dry-cured loin [47,49] and dry-cured ham [30,50,51].

Summarizing, the water, fat, ash, and salt content, as well as pH and water activity, of Pul and ZS dry-cured meat products changed depending on the breed and time of maturing. The count of lactic acid bacteria, as well as the contents of fatty acids, and basic amino acids (arginine, histidine, lysine), shaped the final pH. The high content of aliphatic and catechol amines in ZS products indicated more advanced proteolysis and decarboxylation of ornithine, lysine, and tyrosine, as well as a twofold greater share of heterocyclic amines via catabolism of histidine and tryptophan. The Pul and ZS products differed in the proportion of decanoic, lauric, stearic, and arachidic fatty acids. There were no significant differences in the sum of ω-6 acids, and the only acid slightly differing Pul and ZS products in the ω-3 group was CLA. The compounds that differentiated Pul and ZS products were obtained as a result of microbial activity and lipid oxidation. Volatile compounds shaped a buttery-fruity aroma with caramellic nuances and with the intensity characteristic for native pork meat.

## 4. Conclusions

Ecological production is associated with the high quality of raw materials, additives, health-promoting properties, safety, animal welfare, and sustainable development of agricultural production. The use of raw materials obtained from native breeds is becoming popular, which is associated with cultural heritage and supporting local producers. Based on these results, it can be mentioned that the meat of both native breeds can become the basis of dry-cured meats. The produced hams are of high quality, although the maturing processes in the assessment of many properties differ significantly. The meat of Pulawska and Zlotnicka Spotted breeds is an interesting target for native raw material studies with nonaccelerated metabolism of animal tissues.

## Figures and Tables

**Figure 1 foods-10-01597-f001:**
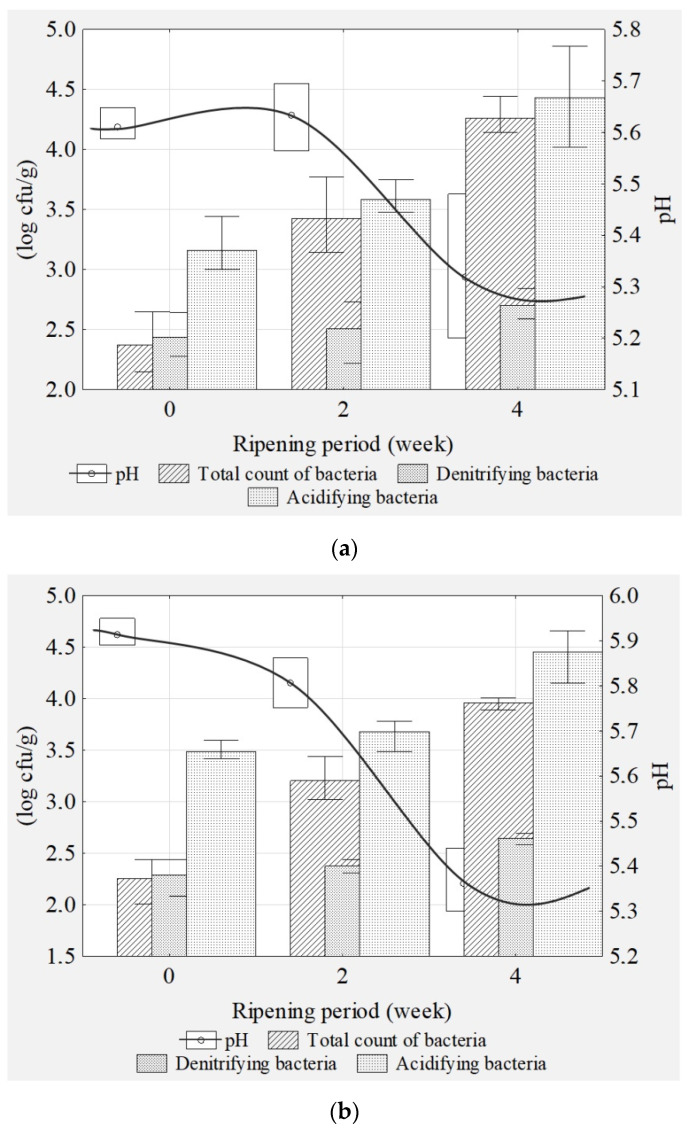
The microbiological characteristics during ripening (mean, standard deviation, *n* = 3) of Pul (**a**) and ZS (**b**) products.

**Table 1 foods-10-01597-t001:** The least-squares means from two-factor analysis of variance of the basic chemical composition and physicochemical characteristics in relation to the type of dry-cured ham and the ripening period.

Characteristics	Product (P)		Ripening Period (Week) (RP)			Interaction		
	Pul	ZS	0	2	4	Source of Variance		
	M	M	M	M	M	P	RP	P × RP
Water (%)	72.15a	71.02b	74.29a	72.19b	69.12c	*	*	*
Protein (%)	18.83a	20.63b	18.19a	18.79a	20.85b	*	*	ns
Fat (%)	4.99a	4.53b	3.43a	4.77b	6.44c	*	*	*
Ash (%)	3.02a	4.04b	1.31a	3.96b	4.55c	*	*	*
Salt (%)	1.89a	1.70b	0.42a	2.47b	2.64c	*	*	*
pH	5.52a	5.69b	5.69a	5.68a	5.33b	*	*	*
a_w_	0.97a	0.96b	0.98a	0.96b	0.95c	*	*	*

Pul—Pulawska; ZS—Zlotnicka Spotted; M—mean; * F values of interaction significant at *p* < 0.05, *n* = 9; ns—not significant; a, b, c—values in rows with different letters differ significantly at *p* < 0.05, *n* = 9.

**Table 2 foods-10-01597-t002:** One-factor analysis of the variance of the free amino-acid content of dry-cured ham.

Product	Pul		ZS		Product	Pul		ZS	
Amino Acid (mg/kg)	M	SD	M	SD	Amino Acid (mg/kg)	M	SD	M	SD
Asparagine	11.47a	0.95	4.02b	0.16	Valine	28.14a	0.69	22.69b	0.89
Serine	34.70a	1.13	32.68b	1.28	Methionine	23.91a	0.59	18.81b	0.73
Glutamine	44.82a	2.82	23.18b	0.91	Lysine	26.76a	1.67	22.92b	0.90
Glycine	23.08a	1.22	17.14b	0.67	Isoleucine	25.63a	0.75	19.89b	0.78
Histidine	25.81a	1.89	20.65b	0.81	Leucine	40.87a	1.15	32.83b	1.28
Arginine	30.04a	1.84	44.33b	1.97	Phenylalanine	31.56a	1.22	34.65b	1.35
Threonine	43.66a	1.55	31.78b	1.24	Free amine groups (g/100 g)				
Alanine	47.45a	1.17	30.45b	1.19	in H_2_O	0.61a	0.06	0.70b	0.00
Proline	30.87a	1.08	21.33b	0.83	in PTA	0.06a	0.00	0.06a	0.01
Tyrosine	33.56a	2.57	30.97b	1.21					

Pul—Pulawska; ZS—Zlotnicka Spotted; PTA—phosphotungstic acid; M—mean; SD—standard deviation; a, b—values in rows with different letters differ significantly at *p* < 0.05, *n* = 9.

**Table 3 foods-10-01597-t003:** One-factor analysis of the variance of the biogenic amine content of dry-cured ham.

Product	Pul		ZS	
Biogenic Amines (mg/kg)	M	SD	M	SD
Tryptamine	4.73a	1.28	9.08b	0.56
Phenylethylamine	1.28a	0.86	0.25a	0.01
Putrescine	3.97a	2.11	23.66b	0.90
Cadaverine	28.53a	2.94	54.31b	3.03
Histamine	87.89a	16.70	163.19b	4.90
Tyramine	11.07a	0.39	57.16b	2.17
Spermine	2.75a	0.19	1.94b	0.07
Spermidine	21.69a	1.04	10.47b	0.43

Pul—Pulawska; ZS—Zlotnicka Spotted; M—mean; SD—standard deviation; a, b—values in rows with different letters differ significantly at *p* < 0.05, *n* = 9.

**Table 4 foods-10-01597-t004:** One-factor analysis of the variance of the fatty-acid content and TBA index of dry-cured ham.

Product	Pul		ZS		Product	Pul		ZS	
Fatty Acids (%)	M	SD	M	SD	Fatty Acids (%)	M	SD	M	SD
10;0	0.12a	0.01	0.08b	0.00	18; 2n-6	8.28a	5.38	7.80a	0.02
12;0	0.09a	0.00	0.06b	0.00	18; 3n-6	0.05a	0.01	0.06a	0.00
14;0	1.24a	0.20	1.04a	0.01	20; 3n-6	0.19a	0.08	0.16a	0.00
16;0	23.14a	1.95	25.91a	0.24	20; 4n-6	1.01a	0.38	1.23a	0.01
17;0	0.14a	0.02	0.12a	0.01	22; 4n-6	0.18a	0.04	0.16a	0.00
18;0	10.09a	0.34	11.19b	0.13	22; 5n-6	0.02a	0.00	0.01a	0.00
20;0	0.09a	0.00	0.13b	0.01					
Σ SFA	34.91		38.53		Σ ω-6	9.73		9.42	
14;1	0.02a	0.01	0.02a	0.00	18; 3n-3	0.44a	0.33	0.37a	0.00
16;1n-7	4.04a	0.92	3.33a	0.02	20; 4n-3	0.07a	0.05	0.06a	0.00
16;1n-9	0.73a	0.01	0.71a	0.03	20; 5n-3	0.04a	0.02	0.06a	0.00
17;1	0.19a	0.03	0.11b	0.01	22; 5n-3	0.16a	0.09	0.16a	0.01
18;1n-7	4.46a	0.85	3.93a	0.08	22; 6n-3	0.08a	0.03	0.07a	0.00
18;1n-9	44.06a	2.99	42.22a	0.28	CLA	0.06a	0.00	0.05b	0.00
20;1	0.58a	0.02	0.61a	0.03	Σ ω-3	0.85		0.77	
Σ MUFA	54.08		50.93		Σ PUFA	10.58		10.19	
					TBA (mg/kg)	0.32a	0.04	0.52b	0.13

Pul—Pulawska; ZS—Zlotnicka Spotted; M—mean; SD—standard deviation; a, b—values in rows with different letters differ significantly at *p* < 0.05, *n* = 9.

**Table 5 foods-10-01597-t005:** Volatile compounds of dry-cured ham—odor impression according to Flavornet; The Good Scents Company; Resconi et al. [48].

Volatile Compound (ng/g)	Ret. Index ZB-5	Ret. Index ZB-Wax	Ret. Time	Product Pul		Product ZS		Formation	Odor Impression
				M	SD	M	SD		
Ethanol (S)	<500	944	1.72	29.23a	0.78	32.53b	7.56	Microb. act. (fermentation)	Alcoholic
Acetone (S)	<500	816	1.84	10.53a	2.56	9.82b	10.12	Microb. act. (fermentation)	Solvent, ethereal, apple
Thiirane, methyl-	596	885	2.39	11.37a	1.28	27.21b	1.98	Spices (garlic)	
Acetic acid (S)	641	1444	2.98	199.47a	9.22	151.31b	15.67	Microb. act. (fermentation)	Pungent, vinegar
3-Methylbutanal	658	916	3.11	10.55a	0.70	9.15b	1.73	Microb. act. (aa catabolism)	Rancid, raw ham-like
1-Hydroxypropan-2-one	679	1291	3.52	5.92a	0.26	4.87b	1.72	Lipid oxidation	Caramellic, burnt
Allyl methyl sulfide	697	953	3.86	9.56a	0.54	9.38a	0.28	Spices (garlic)	Sulfuric, alliaceous
3-Hydroxybutan-2-one	719	1279	4.45	15.96a	6.13	4.81b	0.98	Microb. act. (fermentation)	Buttery, milky, fatty
3-Methylbutan-1-ol	737	1214	4.98	41.77a	0.19	41.55a	1.46	Microb. act. (aa catabolism)	Pungent, whisky-like
Pyridine (S)	757	1181	5.54	16.43a	2.97	10.09b	0.69	Pyrolysis	Fish-like
Toluen (S)	767	1032	6.09	18.40a	1.34	15.19b	1.77	Pyrolysis	Sweet
Hexanal (S)	801	1075	7.82	11.28a	7.24	42.54b	1.63	Lipid oxidation	Aldehydic, fatty, fruity
2-Cyclopenten-1-one	842	1343	10.12	7.21a	2.68	5.89b	0.02	Lipid oxidation	Caramelized sugar
2-Methylcyclopentanone	848	1181	10.55	11.67a	1.31	11.10a	0.69	Lipid oxidation	Meaty, roasted beef-like
Diallyl sulfide	865	1150	11.86	9.83a	0.90	5.60b	1.58	Spices (garlic)	Alliaceous, metallic
2-Furanmethanol	870	1660	12.20	112.18a	2.89	53.21b	5.40	Spices	Burnt
*p*-Xylen	874	1129	12.62	11.21a	0.30	11.38a	3.15	Pyrolysis	Aromatic
Hexan-1-ol	881	1358	13.12	5.57a	1.00	6.72b	1.13	Spices	Freshly mown grass
*o*-Xylene (S)	891	1173	14.16	10.53a	0.36	5.99b	0.73	Pyrolysis	Geranium
2-Cyclopenten-1-one, 2-methyl-	905	1357	15.13	28.15a	1.02	19.66b	2.68	Lipid oxidation	Caramelized sugar
Allyl methyl disulfide	915	1269	15.63	5.93a	2.56	7.25b	5.99	Spices (garlic)	Alliaceous
γ-Butyrolactone	915	1599	15.65	21.77a	1.80	21.30a	3.00	Lipid oxidation	Creamy, caramel
2(*5H*)-Furanone	917	1754	15.79	7.38a	0.56	6.30b	3.73	Lipid oxidation	Buttery
α-Pinene (S)	936	1013	16.75	17.92a	0.28	17.26a	1.17	Spices (pepper)	Camphoraceous, pine
Camphene	950	1053	17.56	8.39a	0.53	6.11b	0.26	Spices	Pungent
2-Furanmethanol, 5-methyl-	963	1728	18.36	9.40a	0.84	4.80b	0.19	Spices	Burnt
2-Cyclopenten-1-one, 3-methyl-	969	1504	18.63	15.48a	1.75	8.70b	0.27	Lipid oxidation	Caramelized sugar
β-Pinene (S)	976	1087	19.17	14.89a	0.44	14.42a	1.39	Spices (pepper)	Piney, eucalyptus
Oct-1-en-3-ol (S)	985	1453	19.65	6.10a	0.62	7.76b	0.08	Spices	Mushroom
β-Myrcene	993	1161	20.20	12.53a	0.64	9.42b	0.39	Spices	Woody, rosy, celery
α-Phellandrene	1001	1152	20.73	9.31a	1.30	7.77b	0.34	Spices (pepper)	Citrus, terpenic
3-Carene	1008	1134	21.02	58.78a	18.71	47.89b	8.01	Spices	Sweet, pungent
*p*-(*o*- or *m*-) Cymene	1026	1261	21.79	13.22a	0.15	10.29b	1.32	Spices	Terpenic, citrus
Limonene (S)	1030	1183	21.98	80.92a	0.74	65.34b	6.47	Spices (pepper)	Menthol, orange
2-Ethylhexanol	1034	1492	22.18	5.68a	0.37	6.02b	0.88	Spices	Fresh floral, fatty
2,3-Dimethyl-2-cyclopenten-1-one	1040	1526	22.47	19.38a	0.39	10.78b	1.29	Lipid oxidation	Caramelized sugar
Phenylacetalaldehyde	1045	1618	22.72	20.47a	3.33	8.80b	1.05	Lipid oxidation	Almond-like
γ-Terpinene	1061	1234	23.48	14.59a	1.08	11.55b	2.58	Spices (pepper)	Terpenic, tropical
Linalool oxide (S)	1077	1439	24.23	8.53a	0.65	4.90b	0.22	Spices	Orange, rose, terpenic
Diallyl disulfide	1081	1487	24.38	107.78a	10.27	67.41b	1.11	Spices (garlic)	Green onion meaty
Guaiacol	1089	1866	24.89	99.79a	5.04	66.44b	3.93	Pyrolysis	Whisky, roasted coffee
Linalool (S)	1100	1542	25.44	222.12a	1.98	144.86b	1.83	Spices	Orange, rose, terpenic
2-Phenylethanol	1114	1923	26.00	6.33a	1.80	4.02b	0.32	Spices	Rose, floral
Camphor	1146	1498	27.29	15.48a	0.93	10.03b	0.58	Spices	Camphoraceous
Creosol	1193	1948	29.32	46.14a	5.35	27.56b	1.76	Pyrolysis	Coal tar
4-Ethyl guaiacol	1281	2034	32.69	15.27a	1.68	8.51b	0.55	Pyrolysis	Whisky, roasted coffee
Eugenol	1364	2176	35.41	73.98a	8.34	43.39b	1.27	Spices	Clove-like, bacon-like
Caryophyllene	1429	1572	37.23	11.83a	0.71	11.09a	3.22	Spices	Pepper-like, citrus

Pul—Pulawska; ZS—Zlotnicka Spotted; M—mean; SD—standard deviation; a, b—values in rows with different letters differ significantly at *p* < 0.05, *n* = 9; (S)—authentic standard of the compound.

## Data Availability

Data available on request from the corresponding author.

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
