# Peer review of "Biochemical Properties Affecting the Nutritional Quality, Safety, and Aroma of Dry-Cured Products Manufactured from Meat of Rare Native Pig Breeds"

_foods, 2021, doi:10.3390/foods10071597_

Round 1

Reviewer 1 Report

* Comments and Suggestions for Authors (will be shown to authors)(Foods 1246052)

It is worth to comment the high effort of the authors to perform different analytical technique to characterize the products. However, there is a need to improvement the manuscript quality and the ideas showed.

English in general is rough to follow and therefore it is recommended to improve it.

  1. The introduction

The introduction could be streamlined to improve its flow. At the moment, it covers many information about the breeds under investigation but it is not clear the information about the product under study. It would benefit from a re-structure that's focused on the dry-cured product and its traditional elaboration. This will help also to focus the results discussion and compare with other research.

Lines 37-40: “… due to the appropriate proportion of water (Which proportion? indicate) to other basic chemical components and sufficient water holding capacity (What is sufficient? Indicate range), is beneficial for manufacturing of regional products of a unique culinary value…

  1. Material and methods

Section 2.1. It is missing additional information about the pigs, what was the slaughter mass? How were these animals feed? How were them slaughter? How many animals were involved in the study? When were the ham tear apart from pigs (after chilling 24 hours or immediately after slaughtering?

Indeed, for those that are not familiarized with the traditional procedure of producing the ham, it is not clear the different steps followed and their order. It would be interesting to include a figure clarifying the dried-product elaboration  and experimental design.

Lines 82-83.  “The wet curing method allowed the penetration of relatively small pieces of meat”.

What do you mean?

Lines: 89-91: “Three batches were produced. Three whole dry cured hams (three Pul and three ZS products) as the replications in each batch and three samples from each ham were analysed.”

The experimental design is poor, low number of replications. The decision of using such a low number of ham has to be justified. A figure to illustrate the experimental design is needed.

Section 2.2.” Sampling”.

Once samples were collected, were they frozen until analysis? If yes, how were them thawed. Were they grind before performing the analysis?

Lines 97-100. ”After separating, the samples for the basic chemical composition (water, protein,7 fat, ash, salt), pH, water activity and the total count of microorganisms as well as the count  of acidifying and denitrifying microflora were estimated as the process of ripening progressed”.

Rewrite

Section 2.3.1. ”Basic chemical composition, physicochemical and microbiological characteristics”.

ISO Procedures are mentioned but not included in the References list. Please include.

Lines 106-17. The water and ash content were determined by drying samples to 106 their stable weight (ISO 1442:2000).

The ISO procedure refers to the measured parameter as moisture content. It is recommended to use the same terminology in the manuscript. Similar indication for “fat” and “salt content”.

Lines 124-125: “Free amino groups were evaluated in water (WNS) and in phosphotungstic acid (PTA) [18]”.

The provided reference did not evaluate what it is indicated in the statement. Please, provide an appropiate reference and also indicate the PTA method how was done.

Lines 125-126. “The WNS fraction (Water Soluble Nitrogen) was used to determine the free 125 amino acids as follows: 5 mL of the water-soluble protein fraction was added 5 mL of 40%...”

 It is not explained how it is the WNS fraction obtained. Add the information.

Lines 132-133 “…with the use of analytical kit ACCQ Tag (Waters, USA).”

Indicate that this kit is a pre-column derivative agent.

Line 136: “and water (60:40, v/v) at a flow rate of 1 Ml min-1, according to the Waters procedure”,

Was the method run under isocratic conditions? If yes, indicate. If not indicate the gradient used.

Lines 138-139: “The biogenic amines analysis was performed using HPLC 138 (16) with modifications:”

Start this assay in a new paragraph. The statement structure is not correct.

Line 140: “…acid (TCA), homogenized for 2 min (1000 rpm) and centrifuged at 14000 x g for 20 min at…

Which system-equipment was used for the homogenization? Vortex? Ultraturrax?.,.

Lines 144-147: “The derivatization process was performed by mixing 1 mL of extract with 1 mL of dansyl  chloride acetone solution (10 mg mL-1) and 0.5 mL of saturated NaHCO3 solution. Incubation was carried out at 40°C for 60 min with occasional shaking. The biogenic amines were extracted twice using 1 mL of diethyl ether for 10 min…”

Change NaHCO3 by NaHCO3. Where ws the incubation done? Oven? Thermoblock? The extraction of the biogenic amines was liquid-liquid extraction at room temperature? Indicate

Line 148 : “acetonitryle” Incorrect spelling. Same mistake in line 156.

In general, for the reagents used is missing the supplier and reference and in some cases the concentration.

Section: 2.3.3.Fatty acids profile, oxidative changes and volatile compounds content”.

This are 3 analytical methods that it is correct to be in same section, however, it is recommended to separated them on different paragraphs.

Lines 163-164 “The fatty acid profile was determined by gas chromatography from the fat extracted from the samples of ready-to-eat products [19]”.

It is not clear for me how do you extracted the fat from the ready to-eat products. Briefly explain.

TBA index: It is missing more details about the TBA method.

  1. Results and discussion

This section has lot of information because the has been a high number of analysis ran. However, the authors just focus on indicating the results for their products but do not include discussion with other research from other groups and, when references are added, finding from this papers are not reported. The section can be highly improved.

Section 3.1.

Lines 191-193: “Basic chemical composition, physicochemical and microbiological characteristics despite the removal of the external fat cover of different thickness for both breeds during the boning, Pul and ZS dry-cured products differed in the basic chemical composition”.

First, in material and methods, it is not indicated the fat cover removal. It should be indicated. Second, the explanation is very brief; it should be developed including references.

Table 1. It is not clear If the ripening period results provided on this tale corresponds to Pul or ZS derived products. How is it possible to compare between them if the results per breed are not provided?

Lines 195-197: “Based on the differences in the intramuscular fat content of Pul and ZS products, the energy values were calculated as 157.8 and 149.6 kcal, respectively”. It is better to define the energy density. Indeed, no discussion about the values of this energy obtained is provided in the manuscript.

Line 201-203: “…quality and nutritional value of dry dried products manufactured both by traditional and…”

Dry died is redundant. Indeed, the statement should be deeper explained an supported by references.

Lines 205-207: “Ripening time significantly affected the analyzed total number of microorganisms (P< 0.05), technologically desirable microflora (P< 0.05): lactic acid bacteria and coagulase-negative cocci and the final pH of Pul and ZS products (P< 207 0.05) (Figure 1).”

In material and methods, the 3 microbiological test are defined as total count of microorganisms, acidifying microflora and denitrifying microflora and on this section, the nomenclature is different and should be unified in the whole manuscript, otherwise it could lead misunderstandings.

Figure 1: If possible, it is recommended to improve the figures format.

Lines 241-242: “The high proportion of putrescine, cadaverine and tyramine in Pul products (4.0, 28.5 and 11.1 mg/kg) compared to ZS (23.7, 54.3 and 242 57.2 mg/kg)…”. 

The high proportion of those biogenic amines is in the ZS product and not in Pul ones. Correct.

Lines 247-250. “The content of cadaverine, histamine and tyramine in dry-cured pork shoulder ripened for 3 months remained at a lower level of 7.5, 6.5 and 21.9 mg/kg, respectively [29]. It could have been caused by the overreaction of amines and their further decomposition.”

And what is the hypothesis that may explain the amines’ overreaction? Differences in ripening conditions? Differences in microflora composition?

Lines 259-261. “The dry-cured meat products of Pul and ZS breeds differed in the proportion of selected saturated fatty acids (SFA): decanoic (C10;0), lauric (C12;0), stearic (C:18;0) and ar-260achidic (C20;0) (P< 0.05) (Table 4)”.

 Why do you think this differences occur? May it be that the animals feeding or breed carcass compositions are different? Do these results match with similar studies in other papers?

In tables 2 and 5 there is not super index c. Removed from the table footnote.

  1. Conclusions

It is a summary of the results obtained for both breeds characterization.

The main conclusion is “The high quality of meat from Pulawska and Zlotnicka Spotted breeds made it pos-318 sible to use for the production of a high-quality, safe, 4-week matured raw dry-cured 319 products. The meat of Pulawska and Zlotnicka Spotted breeds is an interesting target for 320 native raw material studies with non-accelerated metabolism of animal tissues”.

The conclusion did not match with the objective indicated in the introduction. 

Author Response

Introduction

  1. The description of the unique properties of meat raw materials has been completed with the characteristics of raw meat products and the technology of their production.
  2. The values ​​for drip loss and water content have been attached.

Material and methods

Section 2.1.

  1. The information of the number of herds, the feeding method, number of carcasses as well as repetition of slaughter and dry-cured ham production has been completed.
  2. The sentence: ”The wet curing method allowed the penetration of relatively small pieces of meat” has been removed (it was obvious).
  3. The slaughter was carried out three times. 30 obtained carcasses were used each time for 3 batches of ham production..” Information has been completed in the text.

Section 2.2.

  1. The sentences have been filled with terms of distribution to appropriate laboratories immediately after the samples were taken for testing.

Section 2.3.1.-2.3.2.

  1. The PTA methodology has been supplemented and explained.
  2. The sentence from lines 132-133: “…with the use of analytical kit ACCQ Tag (Waters, USA)” has been corrected.
  3. The information from line 136: “and water (60:40, v/v) at a flow rate of 1 Ml min-1, according to the Waters procedure” has been filled by gradient conditions.
  4. The sentence from line 140: “…acid (TCA), homogenized for 2 min (1000 rpm) and centrifuged at 14000 x g for 20 min at…” has been supplemented (Vortex).
  5. The sentences from lines 144-147: “The derivatization process was performed by mixing 1 mL of extract with 1 mL of dansyl chloride acetone solution (10 mg mL-1)…” as well as line 148 and 156: “acetonitryle” have been corrected and supplemented.
  6. The standards of biogenic amines have been written.

Section 2.3.3.

  1. As suggested: „This are 3 analytical methods that it is correct to be in same section, however, it is recommended to separated them on different paragraphs” the text has been separated.
  2. The information from lines 163-164: “The fatty acid profile was determined by gas chromatography…” was corrected.
  3. The TBA method has been explained.

Results and discussion

Section 3.1.

  1. In response to a question about lines 191-193: The removal of the fat cover is a technological process that allows the ham to be separated (carcass cutting in a slaughterhouse). Ham is the raw material for the production of cured meat or culinary processing. For this reason, the process of removing the fat cover was not separately described in the research methodology.
  2. In response to a question about Table 1: The results of both breeds' products, collected over all maturation periods, were averaged and presented for comparison in the first two columns.
  3. In response to a question about lines 195-197: The caloric value supplied with Pul and ZS products has been calculated for consumers looking for products with a low energy value.
  4. The sentence from line 201-203: “…quality and nutritional value of dry dried products manufactured both by traditional and…” was corrected.
  5. The sentence from lines 205-207: “Ripening time significantly affected the analyzed total number of microorganisms...” was corrected.
  6. The sentence from lines 241-242: “The high proportion of putrescine, cadaverine and tyramine in Pul products (4.0, 28.5 and 11.1 mg/kg) compared to ZS (23.7, 54.3 and 242 57.2 mg/kg)…” was corrected and completed. The ratio of protein to fat and a slightly different water content could create an environment more or less conducive to the decarboxylation of amino acids by bacteria.
  7. The information from lines 259-261: “The dry-cured meat products of Pul and ZS breeds differed in the proportion of selected saturated fatty acids....” were completed with literature reports.
  8. The footnotes of tables 2-5 were corrected.

Conclusions

The aim of the study was detailed.

Reviewer 2 Report

foods-1246052 Biochemical properties affecting the nutritional quality, safety and aroma of dry-cured products manufactured from meat of rare native pig bread

The manuscript is based on sound expertimental design, and the presentation of finding is well- organized. Followings are some suggestions to improve the quality of manuscript. Once revise, I am ok with accepting the manuscript.

L78-81 Please add little more information about % calculation. what was the denominator of this % ?

Please add detailed explanation of Fig 1. in the results and discussion section

Section 3.3 Fatty acid profile, oxidative changes and volatile compounds content

  • Please add discussions on this section. Has any studies reported the similar results as your finding? Please cite and dicuss in this section

Author Response

Material and methods

Section 2.1.

  1. The information from lines 78-81 have been filled.

Results and discussion

Section 3.1.

  1. The chapter has been supplemented with information on cenobiotic microflora exchange.

Section 3.3.

  1. The chapter has been supplemented with literature information on fatty acids, TBA index and volatile compounds.

Round 2

Reviewer 1 Report

The authors have improved the manuscript according to some of the previous revision report indications, however the manuscript still needs to be English polished to be considered as a proper one. 

 I still miss what is the scientific relevance of this study (it is very descriptive). The conclusions section is a "summary" of the results. Note that in the objective authors speak of the biochemical processes that influence "safety". Are Biogenic Amine Levels Safe? Which are the allowed levels set by the regulations? I say this because in lines 301-313 they only compare the levels of amines between the 2 breeds, but not with the levels of other similar meat products and with those considered "safe". The authors should consider the following papers: 

  • Landeta et al Meat Science 2007; 77(4): 556-561 
  • Lorenzo et al. Meat Science 2007; 77(2): 287-293 
  • Martuscelli et al. Food Chemistry 2009; 116(4): 955-962 
  • Stadnik & Dolatowski. Meat Science 2012; 91(3): 374-377 
  • Virgili et al. LWT 2007; 40(5): 871-878 

Introduction must be streamlines to improve its flow. Not proper linkage in between paragraphs. 

 The experimental design is not clear. A figure to illustrate the experimental design is needed. 

 The analytical ISO procedures need to be included in the References list. 

 The conclusions should not summarize the results. The conclusions provide a clear interpretation of the findings of research in a way that stresses the significance of study.

Author Response

The thematic profile of the chapter Conclusions has been changed as well as experimental design has been accurately depicted in the text. Information on the safety of biogenic amines has been entered into the manuscript with the literature sources recommended by the Reviewer. The analytical ISO procedures have been included in the References list.